

# A 15-year record (2001-2015) of the ratio of nitrate to non-seasalt sulfate in precipitation over East Asia

Syuichi Itahashi[1], Keiya Yumimoto[2], Itsushi Uno[2], Hiroshi Hayami[1], Shin-ichi Fujita[1], Yuepeng Pan[3], Yuesi Wang[3]

[1] Environmental Science Research Laboratory, Central Research Institute of Electric Power Industry (CRIEPI), 1646 Abiko, Abiko, Chiba 270–1194, Japan
[2] Research Institute for Applied Mechanics (RIAM), Kyushu University, 6-1 Kasuga Park, Kasuga, Fukuoka 816–8580, Japan
[3] State Key Laboratory of Atmospheric Boundary Layer Physics and Atmospheric Chemistry (LAPC), Institute of
10 Atmospheric Physics (IAP), Chinese Academy of Sciences (CAS), Beijing 100029, China

*Correspondence to*: Syuichi Itahashi (isyuichi@criepi.denken.or.jp)

**Abstract.** Acidifying species in precipitation can cause severe impacts on ecosystem. The chemical concentration of precipitation is directly related to the precipitation amount, so it is partly difficult to identify the long-term variation from precipitation concentration. The ratio of nitrate ($NO_3^-$) to non-seasalt sulfate ($nss\text{-}SO_4^{2-}$) concentration in precipitation on an

15 equivalent basis (hereinafter, *Ratio*) will be a useful index. To identify the long-term record of acidifying species in precipitation over East Asia, where is the highest emission region in the world, we have compiled the ground-based observations of the chemical concentration of precipitation over China, Korea, and Japan from 2001 to 2015 based on the Acid Deposition Monitoring Network in East Asia (EANET). The period was partly limited but other monitoring data in Japan, southern China, and northern China around Beijing were additionally utilized. The analyzed period was categorized

into three phases: Phase I (2001–2005), Phase II (2006–2010), and Phase III (2011–2015). The behavior of $NO_3^-$ and $nss\text{-}SO_4^{2-}$ concentration, and hence *Ratio* in precipitation will be related to these precursors. The anthropogenic NOx and $SO_2$ emission amount, and $NOx/SO_2$ emission ratio are analyzed. Further, satellite observations of $NO_2$ and $SO_2$ column density to capture the variation in emission was applied. We found that the long-term trend of $NO_3^-$ concentration in precipitation was not related to the variation in NOx emission and the $NO_2$ column. In comparison, the $nss\text{-}SO_4^{2-}$ concentration in

precipitation over China, Korea, and Japan was partly connected to the changes in $SO_2$ emission from China, but the trends were not significant. The long-term trend of *Ratio* over China, Korea, and Japan were nearly flat during Phase I, increasing significantly during Phase II, and almost flat again during Phase III. These variations of *Ratio* in East Asia clearly corresponded to the $NOx/SO_2$ emission ratio and the $NO_2/SO_2$ column ratio in China. The first flat trend during Phase I was due to both increases in NOx and $SO_2$ emissions in China, the significant increasing trend during Phase II was triggered by

the increase in NOx emission and decrease in $SO_2$ emission in China, and the returned flat trend during Phase III was caused by both declines in NOx and $SO_2$ emissions in China. This suggests that China's emission has a significant impact not only on China but also on downwind precipitation chemistry during the analyzed 15–year period of 2001–2015. In terms of wet depositions, the $NO_3^-$ wet deposition amount over China, Korea, and Japan has not changed dramatically, but the $nss\text{-}SO_4^{2-}$





wet deposition amount declined over China, Korea, and Japan from Phase II to III. These declines were caused by a strong decrease in nss-SO$_4^{2-}$ concentration in precipitation accompanied by a reduction in SO$_2$ emission from China, which counteracted an increase in precipitation amount. It was indicated the decision on the acidity of precipitation would be shift from sulfur to nitrogen.

## 1 Introduction

Accompanying the recent acceleration of anthropogenic emissions in Asia, atmospheric deposition on Asia has been highly focused upon worldwide (Vet et al., 2014). In Japan, which is located in the downwind region of Asian continent, it was revealed that the total wet and dry deposition amounts have surpassed the level of those from both the Clean Air Status and
10 Trends Networks (CASTNET) in the US and the European Monitoring and Evaluation Programme (EMEP) in Europe (Endo et al., 2011; Ban et al., 2016). In terms of the wet deposition amount of nitrogen and sulfur compounds over Japan, the influence of anthropogenic emission from China has been determined via chemical transport model simulation (Kuribayashi et al., 2012; Kajino et al., 2011; 2013; Morino et al., 2011).

The amount of wet deposition is affected by the amount of precipitation. For this reason, the ratio of nitrate (NO$_3^-$)
concentration to non-seasalt sulfate (nss-SO$_4^{2-}$) concentration in precipitation on an equivalence basis, hereafter referred to as *Ratio*, is useful for evaluating the relative contributions of nitrogen and sulfur to the acidity of precipitation. *Ratio* in Japan was about 0.41 between 1987 and 1990 (Fujita et al., 2003). Over western Japan, *Ratio* increased substantially by 0.09–0.17 between 1987 and 1996, reaching about 0.5 in the late 1990s. This corresponded to a large increase in NOx emission compared with SO$_2$ emission across East Asia (Takahashi and Fujita, 2000). In our previous studies (Itahashi et al., 2014a,
2015), we showed that the *Ratio* trend in precipitation over Japan remained flat at around 0.5–0.6 between 2000 and 2005, with a subsequent increase to 0.6–0.7 between 2006 and 2011. During this period, the NOx/SO$_2$ emission ratio in Japan had been constantly decreasing, and changes in *Ratio* closely followed the changes in the NOx/SO$_2$ emission ratio in China. A significant increase in the *Ratio* trend between 2006 and 2011 were also found over both China and Korea. It was clarified a correlation between *Ratio* in China, Korea, and Japan and the NOx/SO$_2$ emission ratio in China. A sensitivity simulation via
a regional chemical transport model regarding Chinese anthropogenic emissions indicated that the important attribution of NOx emission near source region and the higher impact of SO$_2$ emission over downwind region. Identifying the importance of acidity via nitrogen and sulfur is critically needed to mitigate ecosystem effects, such as soil acidification (Zhao et al., 2009) and surface water acidification (Yamashita et al., 2016). Similar studies that used *Ratio* (or the inverse of *Ratio*, defined by the ratio of nss-SO$_4^{2-}$ to NO$_3^-$ concentration in precipitation) were conducted in northern China (Wang et al.,
2012), Nanjing (Tu et al., 2005), southeastern China (Cui et al., 2014), and the Pearl River Delta region (Lu et al., 2015).



We next consider recent changes of *Ratio* in East Asia according to the latest changes in anthropogenic emissions. In our previous studies (Itahashi et al., 2014a, 2015), which considered data for 2000–2011, emission from China was shown to transition from a trend of continuous increase. NOx emission in China has been rising steadily (Kurokawa et al., 2013; Itahashi et al, 2014b), whereas $SO_2$ emission in China peaked in 2005–2006 and subsequently declined due to the

introduction of flue-gas desulfurization systems in China's 11th Five-Year Plan (2006–2010) (Kurokawa et al., 2013; Itahashi et al., 2012a). Recently, it has been reported that NOx emission in China declined after 2011–2012 (Irie et al., 2016; Xia et al., 2016; Krotkov et al., 2016; van der A et al., 2017). To determine the cause of this change, we revisited and updated the analysis of precipitation observation data in relation to emission variation over East Asia. We have complied the precipitation observation on 15 years during the period between 2001 and 2015. Considering the drastic change in emission

from China, the analyzed period was divided into three parts: Phase I (2001–2005), Phase II (2006–2010), and Phase III (2011–2015) in this study. This study renovates our previous studies (Itahashi et al., 2014a; 2015) on the following two points. First, this study incorporates the inclusion of additional data over southern China and northern China around Beijing. The ground-based observation network in East Asia does not cover the northern China region, which is characterized by a large urban population and related anthropogenic emissions, and therefore atmospheric concentration and depositions were

centered (e.g., Pan et al., 2015). The approach taken here will further promote our understanding of precipitation chemistry for all of China. Last, this study uses satellite observations of $NO_2$ and $SO_2$ column density as a proxy to estimate the latest emission changes. This can further enhance our knowledge of the current emission status. Analyses for long-term precipitation chemistry are still limited. For example, precipitation chemistry over periods of longer than 10 years has been reported for Guangzhou during 1983–2010 (Fang et al., 2013), Shenzhen during 1986–2006 (Huang et al., 2008), Lijiang

during 1989–2006 (Zhang et al., 2012), and Nanjing during 1992–2003 (Tu et al., 2005). This study's long-term, 15-year analysis will provide a comprehensive view of the precipitation chemistry over East Asia.

   This paper is structured as follows. Section 2 introduces the dataset used in this study, which includes ground-based observations, emission inventories, and satellite observations that were applied as a proxy for emission levels. Section 3 is dedicated to the results and discussion. First, Section 3.1 presents the long-term trends for $NO_3^-$ and nss-$SO_4^{2-}$ in China,

Korea, and Japan. Section 3.2 describes the intensive analysis of *Ratio* and its relation to emissions. Section 3.3 explores the analysis of wet deposition. Finally, Section 4 summarizes this research and outlines future perspectives.

## 2 Dataset

### 2.1 Ground-based observations

The ground-based observations used in this study are spatially mapped in Fig. 1. Detailed information for the observation period, including latitude (°N), longitude (°E), and elevation (meters above sea level (a.s.l)) of each site, is listed in Table 1. The observation dataset of chemical composition in precipitation, which was compiled by the Acid Deposition Monitoring





Network in East Asia (EANET) program, was mainly used in this study. In China, EANET observations were conducted over three areas in southern China (Zhuhai, Xiamen, and Chongqing) and one area in central China (Xi'an), for a total of 10 sites. In Korea, EANET observations from three sites (Cheju, Imsil, and Kanghwa) were available. In Japan, a total of 11 sites were available on EANET. Among the 11 sites, the data at Ogasawara, which is located in the northwest Pacific Ocean

(27.09°N, 142.22°E, 230 m a.s.l), contains data for chemical composition in precipitation that are under the detection limit, which is 11.8% for $NO_3^-$ and 7.7% for nss-$SO_4^{2-}$. Therefore, the data from Ogasawara were excluded in this study. Samples were collected by a wet-only sampler at a daily interval, except for Banryu (sampled weekly during 2001–2015) and Ijira (sampled weekly in 2001). Concentrations of $NO_3^-$ and $SO_4^{2-}$ in precipitation were determined by ion chromatography and qualified via ion balance and conductivity agreement. The completeness of the data was determined from precipitation

coverage duration and total precipitation amount (EANET, 2000; 2010). In Japan, two sites (Ryori and Komae) were included for use in the study. Under the Global Atmosphere Watch (GAW) program of the World Meteorological Organization (WMO), the Japan Meteorological Agency (JMA) has been conducting observations of atmospheric concentration and deposition at the northern, remote Ryori site since 1976; observation of deposition at Ryori ended in 2011. Samples were collected by a wet-only sampler at a daily interval at Ryori. The Central Research Institute of Electric Power

Industry (CRIEPI) conducted continuous monitoring of precipitation chemistry. The CRIEPI data obtained at Komae, which is located near Tokyo, span from 1987 to the present (Fujita et al., 2000). As we have more fully described in our previous studies (Itahashi et al., 2014a; 2015), CRIEPI monitoring was also conducted at Goto Island, which is located at the western edge of Japan. However, the period of coverage was only from February 2000 to April 2003, so we excluded the Goto data from this study. Samples were collected by a wet-only sampler at 10-day intervals in Komae. For the EANET, JMA, and

CRIEPI monthly mean datasets, nss-$SO_4^{2-}$ concentration (in mol/l) was calculated from the conservative assumption that sodium ($Na^+$) is a seasalt tracer, using the following equation.

$$\text{nss-}SO_4^{2-} = SO_4^{2-} - 0.06028 \times Na^+. \qquad (1)$$

To analyze the long-term behavior of the dataset over the 15-year period of 2001–2015, outliers in the observation of EANET, JMA, and CRIEPI were carefully examined according to method used in Itahashi et al. (2015). Monthly mean

concentrations of $NO_3^-$, nss-$SO_4^{2-}$, and *Ratio* in precipitation were analyzed using the Smirnov–Grubbs outlier test at each site. In this study, the outlier of *Ratio* was directly checked. In this method, outliers were detected one at a time, assuming that the most probable distribution was an approximately normal distribution. The hypothesis of no outliers is rejected when

$$G = \frac{\max_{i=1,\dots,N}|C_i - \bar{C}|}{s} > \frac{N-1}{\sqrt{N}} \sqrt{\frac{t^2_{(\alpha/2N,\, N-2)}}{N-2+t^2_{(\alpha/2N,\, N-2)}}}, \qquad (2)$$

where $N, \bar{C},$ and $s$ are the number, mean, and standard deviation, respectively, of $NO_3^-$ or nss-$SO_4^{2-}$ concentrations in

precipitation or *Ratio* in precipitation ($C_i$). $t^2_{(\alpha/2N,\, N-2)}$ denotes the critical value of the t-distribution with $(N-2)$ degrees of freedom and a significance level of $(\alpha/2N)$. Outlier detection and removal was iterated until the dataset satisfied the specified significance level of 0.05. Following the above criteria in this study, 4.7%, 3.8%, and 4.4% of $NO_3^-$, nss-$SO_4^{2-}$, and





*Ratio* in precipitation, respectively were discarded from the China data; 5.0%, 3.0%, and 3.0% of these were discarded from the Korea data; and 2.4%, 1.5%, and 1.6% of these were discarded from the Japan data. Finally, the annual mean concentrations of $NO_3^-$, $nss-SO_4^{2-}$, and the annual mean of *Ratio* in precipitation were calculated from the monthly mean data whenever at least 9 months of data were available for a given year at the site.

These observation datasets taken from EANET, JMA, and CRIEPI were essentially the same as those used in our previous studies (Itahashi et al., 2014a; 2015). A limitation of our previous studies was a lack of spatial coverage over northern China, especially around the capital of Beijing, because EANET covered only the area from southern to central China (Fig. 1). It has been recognized that anthropogenic emissions centered over this region (Kurokawa et al., 2013; Li et al., 2017), related atmospheric concentration, and depositions were severe in China (e.g., Pan et al., 2015). To overcome this limitation and

advance our knowledge of precipitation chemistry over the whole of China, we added the following dataset for the chemical concentration of precipitation over China.

The integrated monitoring program on acidification of the Chinese terrestrial system (IMPACTS) was established through a Chinese–Norwegian cooperative project (Larssen et al., 2004; 2006) from 2001 to 2003. Under the IMPACTS program, atmospheric concentration, precipitation composition, and soil, water, and vegetative effects were studied at five forested

sites (LXH, LGC, LCG, CJT, and TSP; refer to Fig. 1 and Table 1) over southern China. In terms of depositions, four measurements (wet-only, bulk, throughfall collected below the tree canopy and below the ground vegetation) were conducted. We used wet-only sampling data in this study. Observations of the chemical concentration of precipitation by wet-only sampler were reported for four sites (LGS, LCG, CJT, and TSP) from 2001 to 2003 and for one site (LXH) from 2002 to 2003. Data for LGS in 2001 were not used due to insufficient coverage.

The precipitation chemistry over northern China will be critically important. The precipitation sample at Beijing Normal University (BNU) in 2003 (Sun et al., 2004; Tang et al., 2005) and the recent work by the State Key Laboratory of Atmospheric Boundary Layer Physics and Atmospheric Chemistry/Institute of Atmospheric Physics/Chinese Academy of Sciences (LAPC/IAP/CAS) from December 2007 to November 2010 (Pan et al., 2012; 2013; Wang et al., 2012) were included in this study. Using the observation framework from BNU, data on a total of 53 rain events were collected at

Beijing. To prevent contamination from dry deposition, the collector surface was covered with a plastic lid. A detailed description of this collection method is provided in Tang et al. (2005). In the recent work conducted by LAPC/IAP/CAS, a three-year observation from December 2007 to November 2010 was conducted at 10 sites around Beijing. Daily rainwater samples were collected using a customized wet/dry automatic collector. The precipitation sensor opened the collection funnel of the cover device when rainfall began. The details of these measurements are described in Pan et al. (2012; 2013) and Wang et al. (2012). In this study, December 2007 to November 2008 was regarded as the year of 2008, December 2008

to November 2009 was regarded as the year of 2009, and December 2009 to November 2010 was regarded as the year of 2010. For the datasets of IMPACTS and LAPC/IAP/CAS, the $nss-SO_4^{2-}$ concentration was calculated with assuming $Na^+$ as a seasalt tracer based on Eq. (1). Due to the lack of information on $Na^+$ within the samples at BNU, $SO_4^{2-}$ was used; however, it was reported that $Na^+$ was not a major ion component of their samples.



### 2.2 Emission inventories

It is predicted that the variation of $NO_3^-$, $SO_4^{2-}$, and hence *Ratio*, will be directly related to the emission of NOx and $SO_2$. We used the following emission inventories in this study. The Regional Emission inventory in ASia (REAS) version 2.1

(Kurokawa et al., 2013), which covers Asia from 2000 to 2008, was the main data used in this study to obtain NOx and $SO_2$ emissions over China, Korea, and Japan. The recent status of Asian anthropogenic emission was assessed by harmonizing different local emission inventories with a mosaic approach named MIX (Li et al., 2017). MIX covers the years 2008 and 2010. The data from MIX were also used to acquire data on NOx and $SO_2$ emissions over China, Korea, and Japan.

The latest country-level status of emissions in China can be found in the work of Xia et al. (2016). Their emissions data

cover the period from 2000 to 2014. The primary case, which analyzed the penetrations of advanced combustors with improved energy efficiency and air pollutant control devices with improved pollutant removal efficiency, was used in this study. In Korea, the National Institute of Environmental Research (NIER) provided the national emissions amounts via the National Air Pollutants Emission Service, and the latest data reported cover 1999 to 2013 (NIER, 2017). In Japan, the Japan Auto-Oil Program (JATOP) provided the five-year interval emission dataset from 1995 (JATOP, 2012a; 2012b). The

datasets for 2005 and 2010 were used in this study.

Using the total amount of NOx and $SO_2$ emissions estimated via inventories over China, Korea, and Japan, the $NOx/SO_2$ emission ratio on mole basis was calculated by assuming NOx as $NO_2$. The behavior of the $NOx/SO_2$ emissions ratio will reveal a correlation to *Ratio* in precipitation.

**2.3 Satellite observations**

The emission inventories had some time lags to be reconciled. For this, we combined the use of satellite observations of the $NO_2$ and $SO_2$ vertical column density to capture the recent status of NOx and $SO_2$. The $NOx/SO_2$ column ratio reflects the $NOx/SO_2$ emission ratio and was effective in characterizing the correspondence with *Ratio* in precipitation. Recently, satellite observations have been widely used as a proxy for emissions data. $NO_2$ column have been used to capture NOx

emissions (e.g., Miyazaki et al., 2012; Mijling et al., 2013; Itahashi et al., 2014b; Han et al., 2015; Irie et al., 2016) and $SO_2$ column for $SO_2$ emissions (e.g., Lee et al., 2011; Li et al., 2010) and/or volcanic eruptions (e.g., Brenot et al., 2014). Several studies have indicated the importance of different technologies to control emissions (e.g., Li et al., 2010; Wang et al., 2015; Krotkov et al., 2016; van der A et al., 2017). For example, the ratio of OMI derived $SO_2/NO_2$ was used to determine the effectiveness of the flue-gas desulfurization devices for power plants in China (Li et al., 2010; Wang et al., 2015).

The $NO_2$ and $SO_2$ column dataset, which was observed by Ozone Monitoring Instruments (OMI) onboard National Aeronautics and Space Administration (NASA) Earth Observing System (EOS) Aura satellite, was used in this study (NASA, 2017). Aura satellite was launched on 15 July 2004 in a sun-synchronous ascending polar orbit with a local equator crossing



time of 13:45±0:15. During the data period, it measured sunlight backscattered from the Earth over a wide range of ultraviolet and visible wavelengths to derive abundances of ozone and other trace gases important for air quality and climate. Science-quality data operations began on 1 October 2004; hence the data from 2005 to 2015 were used to cover our analyzed period of 2001–2015. Retrieval algorithms were based on the products provided by NASA.

In terms of $NO_2$ column, we used a level-3 daily global nitrogen dioxide product (OMNO2d) of the latest version (3.0), which was released in August 2016 and is gridded at a resolution of $0.25° \times 0.25°$ (Krotkov et al., 2013). This product contains the total and tropospheric column for all atmospheric conditions and for sky conditions where cloud fraction is less than 30%. We analyzed the tropospheric column with clouds screened on the condition of cloud fraction less than 30%.

In terms of $SO_2$ column, we used a level-3 daily global sulfur dioxide product (OMSO2e) of the latest version (3.0), which

was released in February 2015 and is gridded at a resolution of $0.25° \times 0.25°$ (Krotkov et al., 2015). The dataset contains the total column of $SO_2$ in the planetary boundary layer. The algorithm used here was based on using principal component analysis as introduced by Li et al. (2013). Cloud fraction, scene number, solar and satellite viewing angles, and row anomaly flags were provided as ancillary parameters. The data filtering of this level-3 dataset was based on excluding rows with any of the following: anomaly flags, radiative cloud fraction greater than 20%, solar zenith angle greater than 70.0°, or scene

number greater than 58 or less than 3. In addition, we adopted the smoothed method to average out the noise levels of $SO_2$ column, following the research of Koukouli et al. (2016) who provided the anthropogenic loading of $SO_2$ over China as obtained from different satellite sensors. This method smoothed the $SO_2$ column assigned to each of the $0.25° \times 0.25°$ grid cells, which were weighted by the $SO_2$ column of the surrounding eight cells. In this process, the negative values were regarded as zero value because our focus was to construct the $NOx/SO_2$ column ratio from satellite observations. We

discarded two data periods associated with volcanic activities from our analysis, as follows. The Sarychev Volcano in the Kuril Islands (48.09°N, 153.20°E) had an explosive eruption that emitted a huge amount of ash and $SO_2$ at altitudes of 10–16 km (Brenot et al., 2014). The data from 14-22 June 2009 included this large amount of $SO_2$ (over 10 D.U.) in and around the analyzed domain, and so they were discarded from the calculation of monthly and annual mean data. An eruption of the Nabro Volcano in Eritrea (13.37°N, 41.07°E) during 12 June–7 July 2011 was also reported. During the night of 12 June

2011, this volcano started to erupt, and on 14 June 2011, it spewed a volcanic plume across the route of many flights over east Africa and the Middle East (Brenot et al., 2014). The data from 15 June to 9 July 2011 were excluded from the calculation of monthly and annual mean data according to the approach detailed in van der A et al. (2017). In Japan, where many activate volcanoes are located, $SO_2$ was continuously emitted at a level that surpassed anthropogenic emissions (e.g., Itahashi et al., 2017a). Due to the difficulties of attempting to separate the effect of volcanic activity, the $SO_2$ column data for

Japan which did not include the two data periods mentioned above was used.

Based on the daily gridded data of $NO_2$ and $SO_2$ column, monthly averages were calculated first and then the annual averages were calculated. In the calculation of annual averaged $NO_2$ and $SO_2$ column, cells with monthly averaged data not available for at least 9 months were regarded as deficient cells to keep the consistency of criteria adopted for the ground-



based observations of $NO_3^-$, $nss\text{-}SO_4^{2-}$, and *Ratio* in precipitation. The $NOx/SO_2$ column ratio was obtained from the annual averaged gridded data of $NO_2$ and $SO_2$ column.

## 3 Results and Discussion

### 3.1 Long-term trend of chemical concentration in precipitation with relation to emissions

The long-term trends of precipitation amount; NOx emission with $NO_2$ column; $NO_3^-$ concentration in precipitation; $SO_2$ emission with $SO_2$ column; and $nss\text{-}SO_4^{2-}$ concentration in precipitation during 2001–2015 over China, Korea, and Japan are shown in Fig. 2. The averaged values and statistical analyses for trends during Phase I (2001–2005), II (2006–2010), and III (2011–2015) are summarized in Table 2. To support our finding of variation in spatial distribution, satellite observations from 2005 to 2015 were used. $NO_2$ and $SO_2$ column were mapped for 2006–2007 (the first half of Phase II), 2010–2011 (the transition from Phase II to Phase III), and 2014–2015 (the latter half of Phase III) in Fig. 3. The annual variations in $NO_2$ and $SO_2$ column, which were based on linear regression analysis during Phases II and III, are also mapped in Fig. 4.

For the treatment of precipitation amount, months where data were insufficient were the same as when applying the Smirnov-Grubbs test for *Ratio* calculation. As is shown in Fig. 2 (a), the year-to-year variation in precipitation amount was found, and the average annual accumulated precipitation amounts were around 1300, 1100, and 1500 mm/year over China, Korea, and Japan, respectively. Statistical analysis revealed that, except for the increasing and decreasing trend over China and Korea during Phase III ($p < 0.05$), there was no clear change in precipitation amount during the analyzed 15 years.

The NOx emissions and satellite observations of $NO_2$ column are shown in Fig. 2 (b). Over China, a emissions increased from 2001 to 2010, peaked in 2011–2012, and decreased after 2012; as was found in other studies (e.g., Irie et al., 2016; Krotkov et al., 2016; van der A et al., 2017). It was also found via spatial distribution that $NO_2$ column above China peaked during the transition from Phase II and III (left part of Fig. 3 (b)) and the contrasting trend of increase/decrease was revealed during Phases II and III (left part of Fig. 4 (a) and (b)). Over China, the NOx emissions of REAS and Xia et al. (2016) and satellite observations of $NO_2$ column were well matched over the long term. Over Korea, NOx emissions obtained from REAS and NIER showed a slight increase on 2003–2004 and were flat after 2008. There was some mismatch of NOx emissions and $NO_2$ column during 2005–2007. On one hand, NOx emissions showed a decreasing trend; on the other hand, $NO_2$ column was almost flat. Over Japan, both NOx emissions and $NO_2$ column revealed a slight decrease during the 15-year period. Overall, the correlation between NOx emissions and $NO_2$ column suggests that satellite observations of $NO_2$ column can serve as a proxy for NOx emissions.

Such variation in NOx emissions should be related to the change in $NO_3^-$ concentration in precipitation. The long-term trend of $NO_3^-$ concentration in precipitation is shown in Fig. 2 (c). $NO_3^-$ concentration in precipitation occurred in descending order in China, Korea, Japan, with concentration levels around 50 µeq/L in China, and 40 µeq/L in Korea. These levels were around twice the level of 15-20 µeq/L in Japan. The temporal variation found in the $NO_3^-$ concentration in precipitation did





not correspond to the NOx emissions variation (Fig. 2 (b)). This was particularly notable for China: although NOx emissions from 2001 to 2010 were revealed to have doubled, the $NO_3^-$ concentration in precipitation had not increased much. The $NO_3^-$ concentration in precipitation over Korea and Japan also did not show dramatic variation. Statistical analysis for $NO_3^-$ concentration in precipitation (Table 2) found not significant overall trend for Phases I–III, with values remaining within −5

to + 3% over China, Korea, Japan, with the exception of a negative trend of −3.4±1.3% ($p < 0.05$) in Japan during Phase II. When comparing EANET observations with additional datasets, IMPACTS sites around south China showed a lower level of $NO_3^-$ concentration in precipitation while BNU and LAPC/IAP/CAS sites around Beijing showed a higher level. These levels reflected the emission intensity of their regions and were almost within one standard deviation of values obtained from the EANET observation network.

$SO_2$ emissions and satellite observations of $SO_2$ column are shown in Fig. 2 (d). Over China, a trend of steady increase was seen for 2001– 2004, with a peak in 2005–2006 and a decrease after 2006; there thus seems to be a five year time-lag prior to the decrease in NOx emissions and $NO_2$ column. $SO_2$ column, which was available from 2005, showed continuous decline over all of China (the center of Fig. 3) and a decreasing trend was calculated by linear regression (the center of Fig. 4 (a) and (b)). In China, the $SO_2$ emissions of REAS and Xia et al. (2016) and $SO_2$ column were well matched with the variation

observed. Over Korea, $SO_2$ emissions obtained from REAS and NIER showed a trend of slight decrease, which matched with $SO_2$ column reduction, but some discrepancies were found for 2007–2008. The temporal variation of $SO_2$ column in Korea was similar to that in China. From the spatial distribution data (Fig. 3), it was implied that the $SO_2$ column over Korea might have some contamination as an effect from the upwind region of China. This is attributed to $SO_2$ column obtained from $SO_2$ PBL products have its center of mass altitude at about 900 m, which would lead to longer transport and longer

lifetime of $SO_2$ at higher altitudes. Therefore, compared with the correlation between NOx emission and $NO_2$ column, it was difficult to clearly capture the relation between $SO_2$ emissions and $SO_2$ column. Over Japan, $SO_2$ emissions exhibited a trend of slight decrease during the 15-year period, whereas a nearly flat trend was found from $SO_2$ column. This is partly related to $SO_2$ column containing $SO_2$ emissions from volcanoes. Hot spots of $SO_2$ column above south Kyushu (a western island of Japan) and central Honshu (near Tokyo), shown in Fig. 3, were related to the location of volcanic activity during this period.

Nss-$SO_4^{2-}$ concentration in precipitation was also arranged in descending order in China, Korea, and Japan, similar to the $NO_3^-$ concentration in precipitation. Nss-$SO_4^{2-}$ concentration in precipitation was 200–400 μeq/L above China, around 60 μeq/L above Korea, and 30 μeq/L above Japan. The level over China was about tenfold that of Japan, and the level over Korea was almost twice that over Japan (Fig. 2 (e)). $SO_2$ emissions from China reached a peak in 2005–2006 (Fig. 2 (d)) and nss-$SO_4^{2-}$ concentration in precipitation above China showed the highest concentration of around 393.0 μeq/L in 2006, with a

decline in 2015. Statistical analysis revealed that nss-$SO_4^{2-}$ concentrations in precipitation above China were around 230 μeq/L during Phase I and II but were 171.5 μeq/L during Phase III (Table 2). Trends calculated by linear regression were 12.7±8.3%/year (not significant) during Phase I, −20.3±8.8%/year (not significant) during Phase II, and −13.6±5.2%/year ($p < 0.05$) during Phase III. It seems that the variation of nss-$SO_4^{2-}$ concentration in precipitation was partly related to $SO_2$ emission changes. In comparing EANET observations with other datasets, IMPACTS sites around south China showed a





lower level of nss-$SO_4^{2-}$ concentration in precipitation, BNU sites showed a higher level, and LAPC/IAP/CAS sites showed almost the same level. These levels were within one standard deviation of means obtained from the EANET observation network, as was the $NO_3^-$ concentration in precipitation. Variation in $SO_2$ emission from Korea and Japan were constant or declined between 2000 and 2008, and nss-$SO_4^{2-}$ concentration in precipitation above Korea and Japan did not exhibit a clear

relation with $SO_2$ emission variations in those countries. In addition, the increasing trend of $+10.0\pm0.8\%$/year ($p < 0.001$) over Korea during Phase I was not related to the $SO_2$ emissions variation in Korea. Relatively high nss-$SO_4^{2-}$ concentrations in precipitation in Japan during 2001–2002 were partly caused by the $SO_2$ emission from Miyakejima volcano in 2000 (Itahashi et al., 2012, 2014a, 2015). With the exception of an increase in Korea during Phase I and the peak in Japan in 2001–2002, nss-$SO_4^{2-}$ concentrations in precipitation above Korea and Japan were high during 2005–2007 and then

decreased. These variations were similar to those over China and seem connected to the variations in $SO_2$ emission from China, though these trends were not significant.

### 3.2 Long-term trend of *Ratio* in precipitation and relation to emission change in China

The long-term variation in precipitation and $NO_3^-$ and nss-$SO_4^{2-}$ concentrations in precipitation did not provide clear or

significant trends during the analyzed 15-year period above China, Korea, and Japan, with a few exceptions. To further clarify the correlation between anthropogenic emission changes and its impact on precipitation chemistry over East Asia, we focus on *Ratio*. The long-term trends in *Ratio* are analyzed for the period 2001–2015 and shown in Fig. 5. The NOx/$SO_2$ emission ratio and $NO_2$/$SO_2$ column ratio were calculated by the value averaged over each country. Over China, both NOx/$SO_2$ emission ratio and $NO_2$/$SO_2$ column ratio were flat during Phase I, sharply increasing during Phase II, and almost

flat during Phase III. The substantial increase during Phase II was caused by an increase in NOx emission and a decrease in $SO_2$ emission (Fig. 2 (b) and (d)), as discussed in Itahashi et al. (2015). It was interesting to find that the trends of the NOx/$SO_2$ emission ratio and $NO_2$/$SO_2$ column ratio were again flat during Phase III. This is because both declined in NOx and $SO_2$ emissions. Based on the spatial mapping of the $NO_2$/$SO_2$ column ratio (right part of Fig. 3), the $NO_2$/$SO_2$ column ratio was substantially lower than the value of 1.0 (red colors in right part of Fig. 3) above China during the first half of

Phase II, and subsequently increased above 1.0 (yellow to green colors in right part of Fig. 3) during the latter half of Phase II and into Phase III. An increase was found over the whole of China during Phase II, but it became only a slight increase over central China and decreases over north and south China during Phase III (right part of Fig. 4 (a) and (b)). Such changes in NOx/$SO_2$ emission ratio were well correlated with the variation in *Ratio* over China. *Ratio* was almost 0.3 during Phase I and subsequently increased to 0.5 during Phase II, with a trend of $+14.8\pm1.9\%$/year ($p < 0.001$) and around 0.4–0.6 during

Phase III with a trend of $+10.1\pm3.8\%$/year ($p < 0.05$). In addition to the variation of *Ratio* found from EANET observations, other observations through IMPACTS, BNU, and LAPC/IAP/CAS also exhibited similar behavior for *Ratio*. From the IMPACTS dataset, which covered southern China with high data intensely during 2001–2003, it was clarified that *Ratio* in China during Phase I was around 0.2, which was a much lower value when compared with the current (Phase III) status.




Although $NO_3^-$ and $nss\text{-}SO_4^{2-}$ concentrations in precipitation were higher than in the EANET observation dataset, observations at BNU revealed a *Ratio* of 0.31, which was well matched with EANET observation results. A coordinated observation network by LAPC/IAP/CAS, which operated for three years around Beijing, clarified the increasing trend during Phase II from 0.39 in 2008 to 0.59 in 2010. These observations, which were also analyzed in this study, can reinforce the

idea that *Ratio* observed in EANET network can be a representative dataset of China for the precipitation chemistry. In addition, the common variation of *Ratio* over China was revealed by separating the EANET observation sites into 3 sub-categories (EANET, 2000). Among 10 EANET sites in China, total of 6 sites (Xiang Zhou, Xiang Zhou, Hongwen, Guanyinqiao, Haifu, and Shizhan) were classified into urban category, and total of 4 sites (Xiaoping, Jinyunshan, Weishuiyuan, and Jiwozi) were regarded as rural or remote sites. The trend of *Ratio* over 6 urban sites were -0.0±3.4%/year

(not significant) during Phase I, +15.0±3.1%/year ($p < 0.01$) during Phase II, and +4.3±2.4%/year (not significant) during Phase III, and those over 4 rural and remotes sites were 4.0±6.5%/year (not significant) during Phase I, +11.2±2.0%/year ($p < 0.01$) during Phase II, and +15.6±11.3%/year (not significant) during Phase III. The increasing trend during Phase III were found in rural and remote sites, however, this trend was not significant. Both urban and rural/remote sites showed the significant increasing trends only during Phase II.

In our previous studies (Itahashi et al., 2014a; 2015), we have highlighted the impact of $NOx/SO_2$ emission ratio in China on the *Ratio* over downwind countries, including Korea and Japan. $NOx/SO_2$ emissions and $NO_2/SO_2$ column ratio in Korea showed complex variation. These increased during Phase I, stayed almost flat during Phase II, and slightly increased or remained flat during Phase III. It should be noted that $NOx/SO_2$ emission ratio and $NO_2/SO_2$ column ratio were not correlated in some parts of Korea. In particular, a flat trend was seen for $NOx/SO_2$ emission ratio from 2006 to 2007 with a

decline of $NO_2/SO_2$ column ratio; later, there was a flat trend for the $NOx/SO_2$ emission ratio from 2010 to 2011 with an increase of the $NO_2/SO_2$ column ratio. These discrepancies are attributed to complications from using $SO_2$ column as a proxy for $SO_2$ emissions over Korea. *Ratio* over Korea was almost 0.6 during Phase I with a trend of -4.9±1.9%/year ($p < 0.05$). This subsequently increased to 0.8 during Phase II with a trend of +13.6±4.7%/year ($p < 0.05$) and around 0.8–1.0 during Phase III with a trend of +3.9±2.6%/year (not significant). Considering the $NOx/SO_2$ emission ratio during Phase II, it can be

suggested that the variation of *Ratio* in Korea may be connected to the variation of *Ratio* in China.

In Japan, both the $NOx/SO_2$ emission ratio and $NO_2/SO_2$ column ratio expressed declining trends during the 15-year period. However, the long-term variation of *Ratio* did not show a decreasing trend. Statistical analysis showed a *Ratio* over Japan of almost 0.6 during Phase I with a trend of –0.3±1.9%/year (not significant), a subsequent increase to 0.7 during Phase II with a trend of +3.6±1.3%/year ($p < 0.05$), and a level of around 0.7 during Phase III with a trend of +2.5±1.3%/year (not

significant). Considering the continuous decline of $NOx/SO_2$ emission ratio during the analyzed 15 years, the variation of *Ratio* in Japan seems connected to the variation of *Ratio* in China.

The relations between *Ratio* over China and *Ratio* over Korea or Japan are displayed in a scatter-plot (Fig. 6). The data clearly illustrate that the *Ratio* over China and Korea and the *Ratio* over China and Japan increase over time. Along with the variation in the $NOx/SO_2$ emission ratio and $NO_2/SO_2$ column ratio in China, *Ratio* in precipitation over China was flat, then



increased, and then returned to flat during the 15-year period. The variation over China accompanied variations over Korea and Japan with correlation coefficients of 0.84 and 0.81, respectively ($p < 0.001$). Through these results, the impact of emission change in China and the variation of precipitation chemistry at the regional scale for East Asia was revealed.

**3.3 Long-term trend of wet deposition amount**

Finally, we focused on wet deposition amounts, which were calculated by multiplying the chemical concentration of precipitation with precipitation amount. Monthly data were checked via an outlier test for the chemical concentration of precipitation, and the annual accumulated wet deposition amounts were computed for years having at least 9 months of coverage. The long-term temporal variation of $NO_3^-$ and $nss-SO_4^{2-}$ wet deposition amount over China, Korea, and Japan are

10 shown in Fig. 7. Statistical analysis of averaged value and trends during Phases I, II, and III are listed in Table 4. $NO_3^-$ wet deposition amounts were approximately 6, 4, and 3 kg-N/ha over China, Korea, and Japan, respectively, and the year-to-year variation was large. Regarding the statistical analysis for $NO_3^-$ wet deposition amount, there were no significant trends for China, Korea, and Japan during each phase (Fig. 7 (a)). Regarding reactive nitrogen (Nr) deposition, the threshold value of 10 kg-N/ha has been used (e.g., Bleeker et al., 2011). The results of 15-year long-term analysis indicated that wet deposition

of $NO_3^-$ accounts more than half in China and about one third in Korea and Japan of this threshold of Nr deposition.

The $nss-SO_4^{2-}$ wet deposition amount exhibited decreasing trends during Phase II and III. The amount over China was around 30 kg-S/ha during Phase I and below 20 kg-S/ha during III, and the amounts over Korea and Japan decreased to about 2 and 1 kg-S/ha, respectively. Based on statistical analysis, $nss-SO_4^{2-}$ wet depositions showed increasing (non-significant) trends during Phase I and decreasing trends during Phases II and III. Decreasing trends over China and Japan

during Phase II and over Korea during Phase III were significant ($p < 0.05$) (Fig. 7 (b)). It should be emphasized that the precipitation amount over China, Korea, and Japan during Phase II and over China during Phase III increased. Our results indicate that the decreasing trends seen in the $nss-SO_4^{2-}$ wet depositions were caused by a strong decline in the $nss-SO_4^{2-}$ concentration in precipitation, which counteracted an increase in precipitation amount. The reduction of $SO_2$ emissions over Korea and Japan might partly contribute to the decline of the $nss-SO_4^{2-}$ concentration in precipitation. However, taking into

account the correlation between precipitation chemistry over East Asia and emission change in China, the $SO_2$ emission reduction in China after 2005–2006 strongly impacted both local deposition and downwind deposition.

To consider the wet deposition impact from $NO_3^-$ and $nss-SO_4^{2-}$, the sum of wet depositions is shown in Fig. 7 (c). For the 15-year period of analysis, it has been suggested that $nss-SO_4^{2-}$ wet deposition be reduced in East Asia. The fraction of $NO_3^-$ wet deposition to total wet deposition on the unit of kg/ha was also analyzed. The results indicated that the fraction of $NO_3^-$

wet deposition gradually increased during the analyzed 15-year period over East Asia, indicating that further attention should be paid attention to the deposition of $NO_3^-$ and related Nr species. This has been coincided with previous reports on several cities in China; Nanjing (Tu et al., 2005), Shenzhen (Huang et al., 2008), Guangzhou (Fang et al., 2013), Pearl River Delta region (Lu et al., 2015), and Beijing (Wang et al., 2012). Our study reinforced these implications on a whole scale over East



Asia. Excess Nr deposition will result in eutrophication, which normally degrade the biodiversity. It has been demonstrated that the transboundary nitrogen air pollution and deposition were found over East Asia (Itahashi et al., 2016, 2017b). To reduce the impact, further understanding for $NH_3$ emission will have a key role to tackle the multipollutant control strategy (Zhao et al., 2009). In China, the pH of precipitation has not been lower compared to Korea and Japan due to the high
buffering species (Duan et al., 2016); therefore, including base cations should be taken for the forthcoming analysis.

### 4 Summary and Future Perspectives

This study analyzed the chemistry of precipitation in East Asia during 2001–2015, especially focusing on the behavior of *Ratio*, which is defined as $NO_3^-$/nss-$SO_4^{2-}$ concentration in precipitation. The monitoring networks over East Asia were
initially used, and the study was augmented through intense observation campaigns over southern and northern China. The application of the $NO_2$ and $SO_2$ column density satellite observations as a proxy for NOx and $SO_2$ emission was also an update from our previous studies (Itahashi et al., 2014a; 2015). The results for $NO_3^-$ concentration in precipitation suggested that there is no clear relation between NOx emission and correspondence among China, Korea, and Japan. In comparison, nss-$SO_4^{2-}$ concentration in precipitation was partly correlated to the $SO_2$ emission change in China, and a corresponding (but
non-significant) decline in Korea and Japan was also found. The analysis for *Ratio* clarified the trend (flat followed by increase followed by flat) during the 15-year analysis of China, Korea, and Japan, and this variation was correlated to the NOx/$SO_2$ emission ratio and the $NO_2$/$SO_2$ column ratio in China. First flat trend was due to the both increase of NOx and $SO_2$ emission in China, subsequent increase was caused by the increase of NOx emission and the decrease of $SO_2$ emission in China, and returned flat trend was due to the both decrease of NOx and $SO_2$ emission in China. Due to the confounding
impact that the upwind region imposes, it is difficult to use the $SO_2$ column as an accurate proxy of $SO_2$ emissions in Korea. In spite of the increasing trends of precipitation amount, decreasing trends for nss-$SO_4^{2-}$ wet deposition amounts over China, Korea, and Japan were seen after 2005–2006 and matched reductions in $SO_2$ emissions from China. Both nitrogen and sulfur compounds pose acidification risks to ecosystems through atmospheric deposition processes. During the 15 years covered by the study, it has been suggested that sulfur wet deposition in East Asia be reduced. Such reduction seems to be important for
the future as well, and further attention are required for nitrogen. Although this study was focused on wet deposition, synergetic analysis with dry deposition processes is also needed to understand impact of total wet and dry deposition on East Asian ecosystems.

### Acknowledgement

The authors thank EANET and JMA for providing wet deposition measurement data. The authors also acknowledge NASA for providing OMI satellite data of $NO_2$ and $SO_2$. Syuichi Itahashi acknowledges the support of JSPS KAKENHI (Grant





JP16K21690). Yuepeng Pan acknowledges the National Key Research and Development Program of China (Grants 2017YFC0210101, 2016YFD0800302) and the National Natural Science Foundation of China (Grant 41405144) for financial support.

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



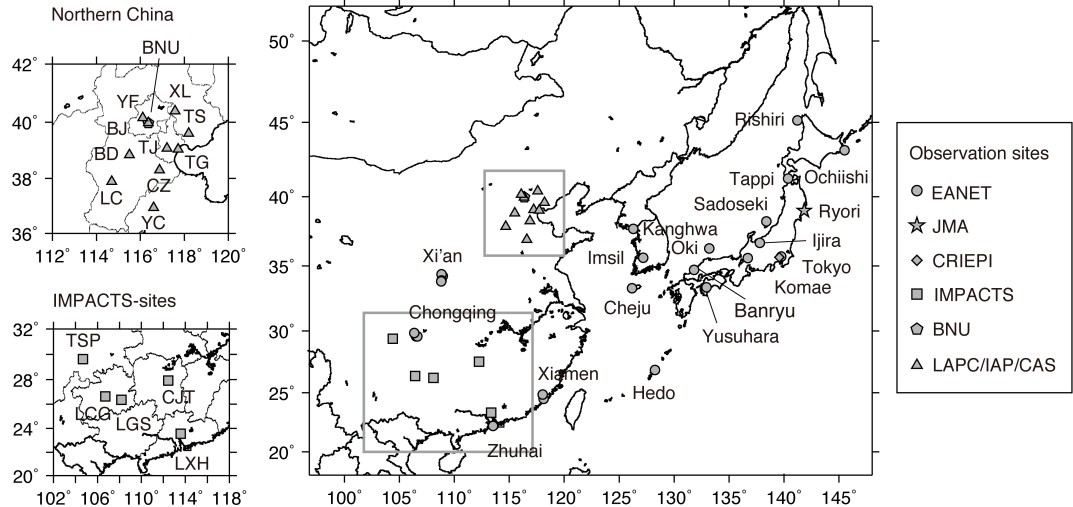

Figure 1: Geographical mapping of observation sites used in this study. Detailed information about each observation site is provided in Table 1.





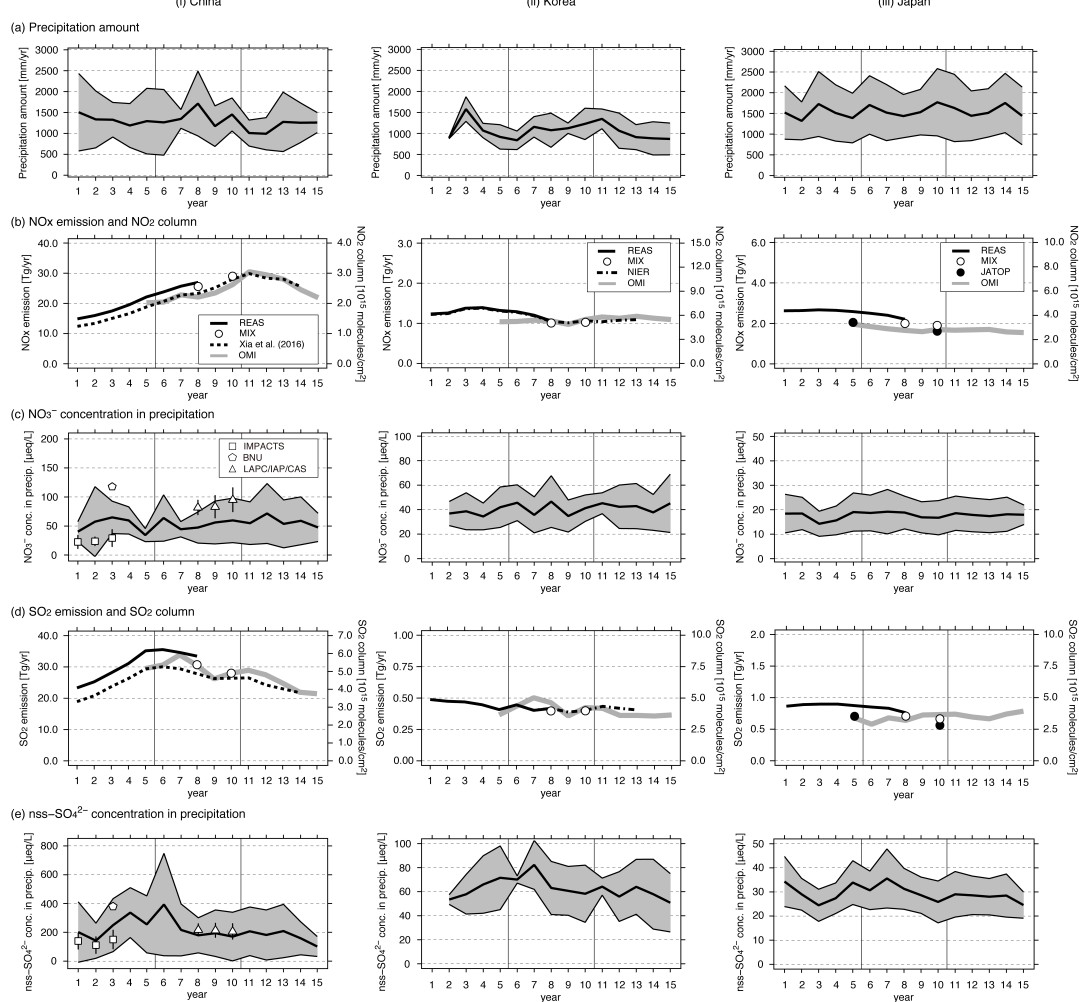

**Figure 2: Long-term temporal variation of (a) precipitation, (b) NOx emission with NO$_2$ column, (c) NO$_3^-$ concentration in precipitation, (d) SO$_2$ emission with SO$_2$ column, and (e) nss-SO$_4^{2-}$ concentration in precipitation during 2001–2015 over (i) China, (ii) Korea, and (iii) Japan. One standard deviation across observation sites is indicated by the shaded areas in (a), (c), and (e). The numbers on the bottom indicate the year since 2001 (e.g., 5 means the year 2005 and 10 means the year 2010).**





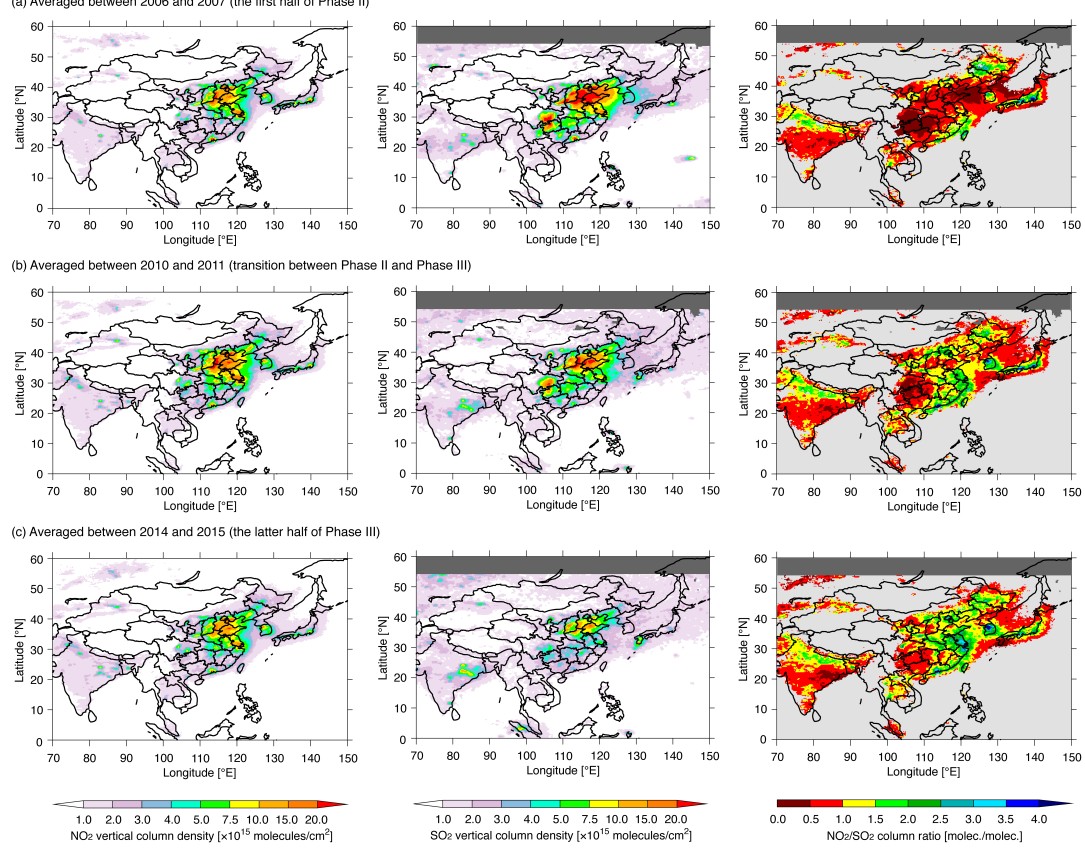

**Figure 3: Satellite observations of (left) NO₂ column, (center) SO₂ column, and (right) NO₂/SO₂ column ratio averaged over (a) 2006–2007 (the first half of Phase II), (b) 2010–2011 (transition between Phase II and Phase III), and (c) 2014–2015 (the latter half**
5 **of Phase III). Dark gray cells indicate places where annual mean could not be calculated. If either NO₂ or SO₂ column is less than 1.0×10¹⁵ molecules/m² in value, this is indicated by a light gray cell for clarity for NO₂/SO₂ column ratio.**



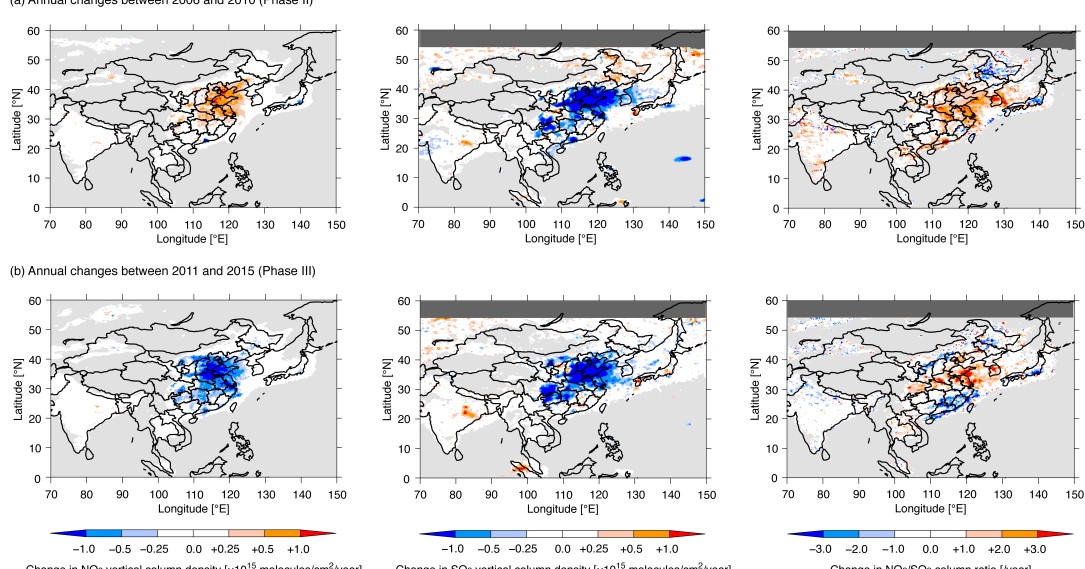

**Figure 4: Annual changes based on the linear regression results of (left) NO₂ column, (center) SO₂ column, and (right) NO₂/SO₂ column ratio during (a) Phase II and (b) Phase III. Dark gray cells indicate areas where annual mean calculation could not be performed. If either NO₂ or SO₂ column are less than 1.0×10¹⁵ molecules/m² in value, this is indicated by a light gray cell for clarity.**



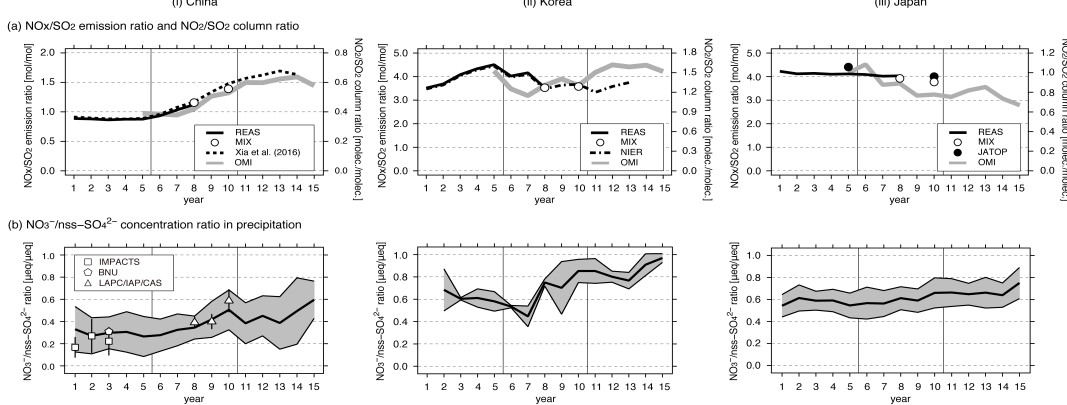

**Figure 5: Long-term temporal variation of (a) NOx/SO$_2$ emission ratio and NO$_2$/SO$_2$ column ratio, and (b) NO$_3^-$/nss-SO$_4^{2-}$ concentration in precipitation (*Ratio*) during 2001–2015 over (i) China, (ii) Korea, and (iii) Japan. One standard deviation across observation sites is indicated by the shaded area in (b). The numbers on the bottom indicate the year since 2001 (e.g., 5 means the year 2005 and 10 means the year 2010).**





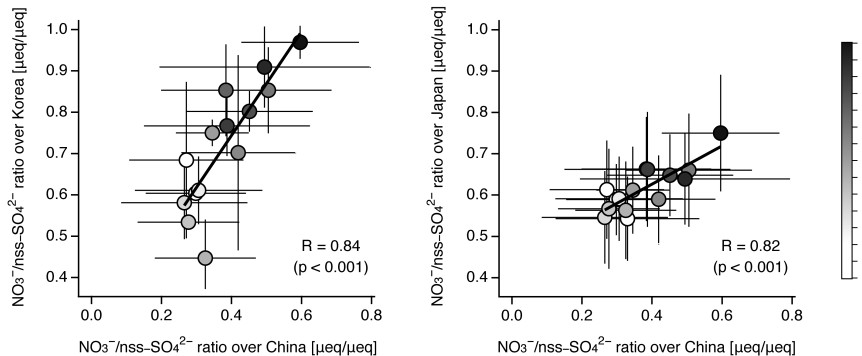

**Figure 6: Scatter-plots of (left) NO$_3^-$/nss-SO$_4^{2-}$ concentration in precipitation (*Ratio*) in China and Korea, and (right) those in China and Japan. Each circle indicates the annual mean *Ratio*, with color indicating the analyzed year.**





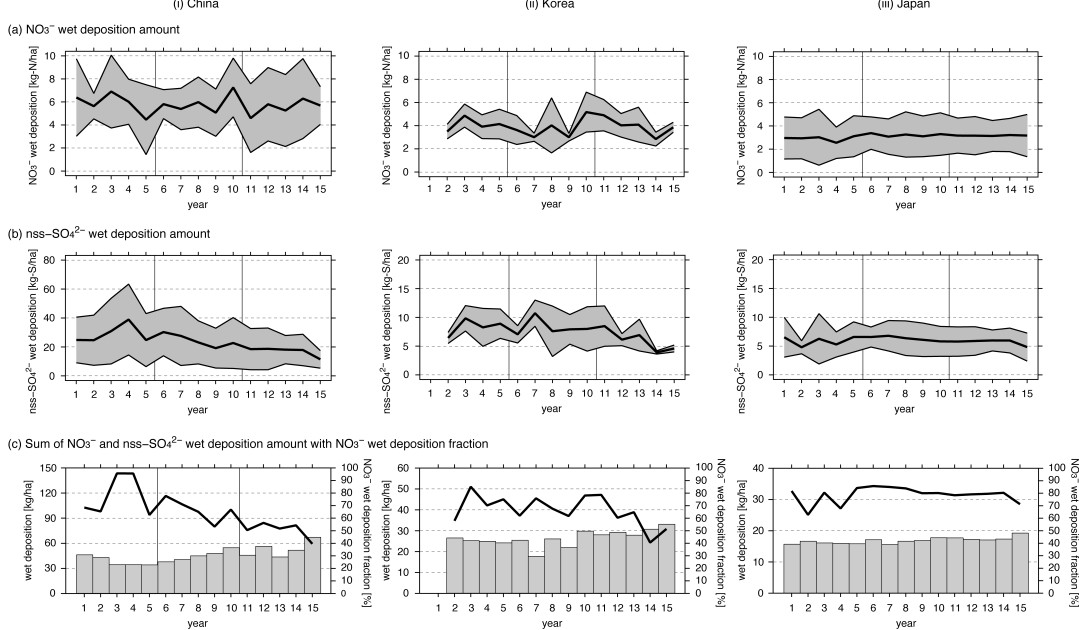

**Figure 7: Long-term temporal variation of (a) NO$_3^-$ wet deposition (kg-N/ha), (b) nss-SO$_4^{2-}$ wet deposition (kg-S/ha), and (c) sum of NO$_3^-$ and nss-SO$_4^{2-}$ wet deposition amount (kg/ha) with NO$_3^-$ fraction during 2001–2015 over (i) China, (ii) Korea, and (iii) Japan.**
5 **One standard deviation across observation sites is indicated by the shaded area in (a) and (b). The numbers on the bottom indicate the year since 2001 (e.g., 5 means the year 2005 and 10 means the year 2010).**





**Table 1. Location of observation sites used in this study.**

|  | Country | Site | Period | Latitude (°N) | Longitude (°E) | Elevation (m a.s.l.) |
|---|---|---|---|---|---|---|
| EANET | China | Zhuhai |  |  |  |  |
|  |  | - Xiang Zhou | 2001–2015 | 22.27 | 113.57 | 40 |
|  |  | - Zhuxiandong | 2008–2015 | 22.20 | 113.52 | 45 |
|  |  | Xiamen |  |  |  |  |
|  |  | - Hongwen | 2001–2015 | 24.47 | 118.13 | 50 |
|  |  | - Xiaoping | 2001–2015 | 24.85 | 118.03 | 686 |
|  |  | Chongqing |  |  |  |  |
|  |  | - Guanyinqiao | 2001–2007 | 29.57 | 106.52 | 262 |
|  |  | - Haifu | 2008–2015 | 29.62 | 106.50 | 317 |
|  |  | - Jinyunshan | 2001–2015 | 29.82 | 106.37 | 800 |
|  |  | Xi'an |  |  |  |  |
|  |  | - Shizhan | 2001–2015 | 34.23 | 108.95 | 400 |
|  |  | - Weishuiyuan | 2001–2006 | 34.37 | 108.85 | 366 |
|  |  | - Jiwozi | 2001–2015 | 33.83 | 108.80 | 1800 |
|  | Korea | Cheju | 2001–2011 | 33.30 | 126.17 | 72 |
|  |  | Imsil | 2001–2011 | 35.60 | 127.18 | - |
|  |  | Kanghwa | 2001–2011 | 37.70 | 126.28 | 150 |
|  | Japan | Hedo | 2001–2015 | 26.87 | 128.25 | 60 |
|  |  | Yusuhara | 2001–2015 | 33.38 | 132.94 | 790 |
|  |  | Banryu | 2001–2015 | 34.68 | 131.80 | 53 |
|  |  | Ijira | 2001–2015 | 35.57 | 136.69 | 140 |
|  |  | Tokyo | 2007–2015 | 35.69 | 139.76 | 26 |
|  |  | Oki | 2001–2015 | 36.29 | 133.19 | 90 |
|  |  | Sadoseki | 2001–2015 | 38.25 | 138.40 | 136 |
|  |  | Tappi | 2001–2015 | 41.25 | 140.35 | 106 |
|  |  | Ochiishi | 2003–2015 | 43.16 | 145.50 | 49 |
|  |  | Rishiri | 2001–2015 | 45.12 | 141.21 | 40 |
| JMA | Japan | Ryori | 2001–2015 | 39.03 | 141.82 | 260 |
| CRIEPI | Japan | Komae | 2001–2015 | 35.64 | 139.58 | 27 |
| IMPACTS | China | Guangdong |  |  |  |  |
|  |  | - Liu Xi He (LXH) | 2002-2003 | 23.55 | 113.58 | 500 |
|  |  | Guizhou |  |  |  |  |
|  |  | - Lei Gong Shan (LGS) | 2002-2003 | 26.37 | 108.18 | 1630-1735 |
|  |  | - Liu Chong Guan (LCG) | 2001-2003 | 26.63 | 106.72 | 1320-1400 |
|  |  | Hunan |  |  |  |  |
|  |  | - Cai Jia Tang (CJT) | 2001-2003 | 27.92 | 112.43 | 450-500 |
|  |  | Chongqing |  |  |  |  |
|  |  | - Tie Shan Ping (TSP) | 2001-2003 | 29.63 | 104.68 | 450-500 |
| BNU | China | Beijing |  |  |  |  |
|  |  | - Beijing Normal University | 2005 | 39.96 | 116.37 | 40 |
| LAPC/IAP/CAS | China | Shandong |  |  |  |  |
|  |  | - Yucheng (YC) | 2008-2010 | 36.85 | 116.55 | 23 |
|  |  | Hebei |  |  |  |  |
|  |  | - Luancheng (LC) | 2008-2010 | 37.89 | 114.69 | 57 |
|  |  | - Cangzhou (CZ) | 2008-2010 | 38.30 | 116.87 | 10 |
|  |  | - Baoding (BD) | 2008-2010 | 38.85 | 115.50 | 21 |
|  |  | - Tangshan (TS) | 2008-2010 | 39.60 | 118.20 | 24 |
|  |  | - Xinglong (XL) | 2008-2010 | 40.38 | 117.57 | 872 |
|  |  | Tianjin |  |  |  |  |
|  |  | - Tanggu (TG) | 2008-2010 | 39.04 | 117.72 | 0 |
|  |  | - Tianjin (TJ) | 2008-2010 | 39.08 | 117.21 | 6 |
|  |  | Beijing |  |  |  |  |
|  |  | - Beijing (BJ) | 2008-2010 | 39.96 | 116.36 | 57 |
|  |  | - Yangfang (YF) | 2008-2010 | 40.15 | 116.10 | 73 |



**Table 2. Statistical analysis of averaged value and trend for precipitation, $NO_3^-$ concentration in precipitation, and nss-$SO_4^{2-}$ concentration in precipitation over China, Korea, and Japan during Phases I, II, and III.**

| | Phase I | | Phase II | | Phase III | |
|---|---|---|---|---|---|---|
| | Mean | Trend | Mean | Trend | Mean | Trend |
| Precipitation [mm/year] | | | | | | |
| China | 1330.2 | −4.3±1.9 | 1390.2 | +1.5±5.4 | 1158.3 | +6.6±2.5[*] |
| Korea | 1114.9 | −3.9±15.4 | 1084.3 | +6.9±3.0 | 1015.1 | −11.2±3.3[*] |
| Japan | 1493.5 | −0.5±3.8 | 1591.7 | +0.9±3.2 | 1555.7 | −0.5±3.1 |
| $NO_3^-$ concentration in precipitation [μmol/L] | | | | | | |
| China | 51.3 | −1.9±9.4 | 54.3 | +0.6±5.5 | 57.3 | −4.7±5.0 |
| Korea | 38.0 | +3.0±4.1 | 40.8 | −2.4±4.7 | 42.7 | −1.1±2.5 |
| Japan | 17.2 | −0.9±4.4 | 18.1 | −3.4±1.3[*] | 18.0 | −0.6±0.8 |
| nss-$SO_4^{2-}$ concentration in precipitation [μmol/L] | | | | | | |
| China | 237.0 | +12.7±8.3 | 230.6 | −20.3±8.8 | 171.5 | −13.6±5.2[*] |
| Korea | 62.2 | +10.0±0.8[***] | 66.9 | −6.8±3.6 | 58.6 | −4.2±2.6 |
| Japan | 29.8 | −0.8±5.2 | 30.4 | −5.4±2.9 | 27.8 | −3.3±1.5 |

Note: Significance for trends is indicated by [*] for $p < 0.05$, [**] for $p < 0.01$, and [***] for $p < 0.001$, and lack of mark indicates no significance. Analysis for Korea during Phase I was for the 2002–2005 period.



**Table 3. Statistical analysis of averaged value and trend for _Ratio_ in precipitation over China, Korea, and Japan during Phases I, II, and III.**

| | Phase I | | Phase II | | Phase III | |
|---|---|---|---|---|---|---|
| | Mean | Trend | Mean | Trend | Mean | Trend |
| _Ratio_ [mol/mol] | | | | | | |
| China | 0.29 | −3.3±2.7% | 0.37 | +14.8±1.9%[***] | 0.46 | +10.1±3.8%[*] |
| Korea | 0.62 | −4.9±1.9%[*] | 0.66 | +13.6±4.7%[*] | 0.86 | +3.9±2.6% |
| Japan | 0.58 | −0.3±1.9% | 0.60 | +3.6±1.3%[*] | 0.67 | +2.5±1.3% |

Note: Significance for trends is shown with [*] for $p < 0.05$, [**] for $p < 0.01$, and [***] for $p < 0.001$, and lack of mark indicates no significance. Analysis for Korea during Phase I was for the 2002–2005 period.



**Table 4. Statistical analysis of averaged value and trend for $NO_3^-$ and nss-$SO_4^{2-}$ wet deposition over China, Korea, and Japan during Phases I, II, and III.**

| | Phase I | | Phase II | | Phase III | |
|---|---|---|---|---|---|---|
| | Mean | Trend | Mean | Trend | Mean | Trend |
| $NO_3^-$ wet deposition [kg-N/ha] | | | | | | |
| China | 5.88 | −5.9±4.6 | 5.90 | +4.4±4.5 | 5.53 | +4.9±3.2 |
| Korea | 4.10 | +2.4±7.5 | 3.76 | +8.2±7.3 | 3.95 | −8.3±4.8 |
| Japan | 2.92 | −0.3±2.7 | 3.23 | −0.4±1.5 | 3.17 | +0.2±0.3 |
| nss-$SO_4^{2-}$ wet deposition [kg-S/ha] | | | | | | |
| China | 28.80 | +4.9±7.4 | 24.57 | −9.6±3.4[*] | 16.90 | −8.8±4.3 |
| Korea | 8.36 | +7.0±8.0 | 8.26 | −1.1±6.3 | 6.01 | −16.7±5.6[*] |
| Japan | 5.88 | +1.0±5.0 | 6.32 | −3.5±0.9[*] | 5.68 | −3.2±2.5 |

Note: Significance for trends is shown with [*] for $p < 0.05$, [**] for $p < 0.01$, and [***] for $p < 0.001$, and lack of mark indicates no significance. Analysis for Korea during Phase I was for the 2002–2005 period.

