# Peer review of "A 15-year record (2001-2015) of the ratio of nitrate to non-seasalt sulfate in precipitation over East Asia"

_Atmospheric Chemistry and Physics, 2017_

## Referee Comment (RC1) · Anonymous Referee #2 · 7 Nov 2017

The long-term trend of NO3- and SO42- concetration in precipitation in China, Korea, and Japan was analyzed based on EANET monitoring data. The results showed certain correlation between China's emission and the wet deposition in East Asia. The manuscript was quite well written. However, the method for trend analysis was not clear introduced. Since there are a lot of anthropogenic sources and other natural sources of Na+ than sea-salt in China, the necessity of sea-salt correction for SO42- concentration in precipitation need be reconsidered. Some detailed comments are:

Page 1, Line 15: The ratio is an index only for relative contribution of SO2 and NOx.

Page 3, Line 21: As I know, a long-term (2001-2013) monitoring of precipitation chem-

istry at one of the IMPACTS site (TSP) was recently reported by Yu et al. (ES&T, 2017).

Page 4, Line 21: I suggest not to do sea-salt correction for SO42-, at least in China.

Page 4, Line 24: Is the monthly mean concentration volume-weighted?

Page 5, Line 4: I can not see the introduction on the method for trend analysis.

Page 5, Line 32: Na+ can not be used as sea-salt tracer, at least in China.

Page 11, Line 7: Double 'Xiang Zhou'.

Page 26, Line 4: Better use equivalence as the unit for the sum.

Page 28 and 29, Table 2 and 3: How to calculate the trend?

---

## Referee Comment (RC2) · W. Aas (Referee) · 16 Nov 2017

This study compare observed trends in in-situ observations of sulfate and nitrate in precipitation, and satellite data of NO2 and SO2 with emission estimates of Sox and Nox in East Asia for the last 15 years. This is an important topic since there has been large changes in the emissions in Asia the last decades. The comparison between satellite data with emission, I think is good, but the comparison with in situ data I find a bit shallow. The spatial variability of the emissions and the wet deposition are large, and the site representatvity is a critical question. Are the sites in question able to show the overall trends for the different regions? For South Korea and Japan it is probably

OK, but in China it is too few sites. The authors are aware of the problem and add campaign data in China, but these are only snapshots in time and cannot be used for assessing trends. I do miss more discussion of this. As I understand it, the authors claim that the ratio of NO3/SO4 is a good parameters for assessing the regional trends (better than the individual concentrations), but this is mainly showing that the chemical regime is changing, which is probably correct, maybe not all over the continent though, and does it help us to understand whether the individual trends of NO3 and SO4 are correct?

A critical point is that the paper is the merging all type of sites. In the EANET network, there are many urban sites and (some of) these may not show representative trends for the region. Even though the authors at page 6 line 8-14 discuss this a bit, and claim they (the Chinese urban sites) show similar trends, it is very few sites and only trends for the ratio is presented. I will suggest to separate urban and regional sites in the study, and even better calculate the trends at the individual sites rather than averaging all the sites, which have totally different siting criteria.

I also find the trend calculations a bit too optimistic since you calculate trends from only five years periods, usually one needs at least 7 years (preferably 10) for calculating trends. You have not described how you calculate trends and the significance, except that it is linear.

More specific comments to the text:

* Page 1. The references Endo et al and Ban et al are mainly studying EANET data and not directly with US and Europe. You should rather use US and European references when comparison. Some examples:

EMEP: Tørseth, K. et al. Introduction to the European Monitoring and Evaluation Programme (EMEP) and observed atmospheric composition change during 1972-2009. Atmospheric Chemistry and Physics 12, 5447-5481, doi:10.5194/acp-12-5447-2012 (2012).

CASTNET: Sickles Ii, J. E. & Shadwick, D. S. Air quality and atmospheric deposition in the eastern US: 20 years of change. Atmos. Chem. Phys. 15, 173-197, doi:10.5194/acp-15-173-2015 (2015).

IMPROVE: Hand, J. L., Schichtel, B. A., Malm, W. C. & Pitchford, M. L. Particulate sulfate ion concentration and $SO_2$ emission trends in the United States from the early 1990s through 2010. Atmos. Chem. Phys. 12, 10353-10365, doi:10.5194/acp-12-10353-2012 (2012).

NADP: Lehmann, C. M. B., Bowersox, V. C., Larson, R. S. & Larson, S. M. Monitoring Long-term Trends in Sulfate and Ammonium in US Precipitation: Results from the National Atmospheric Deposition Program/National Trends Network. Water, Air, & Soil Pollution: Focus 7, 59-66, doi:10.1007/s11267-006-9100-z (2007).

UNECE also has relatively new assessments of trends from Europe and North Americ, which might be relevant as well: http://www.unece.org/index.php?id=42861 and http://www.unece.org/index.php?id=42947

*Page 3 line 13-16. I don't understand what you mean here that "the deposition were centred". with a reference to Pan 2015, who looks at deposition of trace elements in Northern China. I assume the authors are discussion the point I address in the beginning with representativity of the EANET sites, but I don't understand how you can state that "The approach taken here will further promote our understanding of precipitation chemistry for all of China" since the additional sites only cover a short period and cannot be used for trends. If you had looked at one specific year to assess the spatial deposition it would have been different.

*Page 4 line 6. The Ogasawara site was excluded with quite strict criteria. 25% is more appropriate. Further you could have in included the Russian site close to the Korean border, Primorskaya, which is also downwind from the large emission sources in China.

*Page 8 line 13-14: "For the treatment of precipitation amount, months where data were insufficient were the same as when applying the Smirnov-Grubbs test for Ratio calculation." I don't understand this sentence

*Page 8 line 16-17 The sentence "Statistical analysis revealed that, except for the increasing and decreasing trend over China and Korea during Phase III .. there was no clear change in precipitation amount..." is somewhat in contrast to the conclusion later on page 13 line 21: "In spite of the increasing trends of precipitation amount, decreasing trends for nss-SO4 wet deposition amounts over China, Korea, and Japan were seen after 2005–2006". The increase in precipitation amount in China is after 2010 (though the variability is very high) and South Korea has decreasing amount.

*Page 8-9 "The temporal variation found in the NO3 concentration in precipitation did not correspond to the NOx emissions variation". This non-linearity can be due to several factors. E.g.: The non representativity of the sites, the change in atmospheric composition and chemical regimes (changes in base cations and ammonium), oxidation capacity of the atmosphere, all may change the lifetime of NOx and NO3 (course or fine).

*Page 9 line 18. "Contamination". It's a bit misleading word. The satellite measurements are not contaminated they are influence by sources outside Korea (like China)

*Page 9 line 23. Have the volcanic activity changed during the period to influence the trend of the SO2 column data? Should maybe also have been included also in the emission inventories?

*Page 9-10. IMPACTS sites have lower levels compared to EANET since these are mainly regional/rural sites while the BNU site is urban and naturally higher than the average EANET site.

*Page 10: There are some contradicting statements:

1)Over China, NO2/SO2 column ratio were flat during Phase I, sharply increasing during Phase II, and almost flat during Phase III.

2)NOx/SO2 emission ratio were well correlated with the variation in Ratio over China

3)Ratio was almost 0.3 during Phase I and subsequently increased.. during Phase II, with a trend of +14.8±1.9%/year and around 0.4–0.6 during 30 Phase III with a trend of +10.1±3.8%/year (p < 0.05).

This leads to the conclusion on page 11 . "Ratio observed in EANET network can be a representative dataset of China for the precipitation chemistry"

This is not obvious for me

*Page 12 line 31. Nr. Is this reactive nitrogen? It is not defined. Can also be interpret as reduced nitrogen. I assume reactive nitrogen since you at page 13 line 1 states that Nr can cause eutrophication and this means both NO3 and NH4

*Figure 7c. It is not clear how sum NO3 and SO4 is calculated Figure C do not add up from fig a and b, even if changing units. Maybe better to look at equivalent and not mass

*Table 2. It would have been nice to indicate how many sites are included in these calculations, and the time period for the different phases should be included in the table so the reader don't need to search in the text. Is it the same number sites in all the phases? In the table there is no unit (kg/ha pr year?)

---

## Author Comment (AC2) · 5 Jan 2018

We appreciate your kind review. Please see the supplemental file for our response.

Please also note the supplement to this comment:
https://www.atmos-chem-phys-discuss.net/acp-2017-848/acp-2017-848-AC2-supplement.pdf

---

## Author Response (AR1)

Dear Dr. Frank Dentener,

Thank you very much for dealing our manuscript, titled "A 15-year record (2001–2015) of the ratio of nitrate to non-seasalt sulfate in precipitation over East Asia," (#acp-2017-848) which we would like to resubmit for publication in *Atmospheric Chemistry and Physics*.

We would also like to thank the anonymous referee and Dr. Wenche Aas for reading our manuscript and providing helpful comments. We have revised our manuscript according to the reviewers' comments and suggestions. We believe that these revisions address all points raised by the reviewers. Our point-by-point responses are provided below, and revisions are indicated in blue in the revised manuscript.

Sincerely,

Syuichi Itahashi

Anonymous Referee #2

The long-term trend of NO3- and SO42- concetration in precipitation in China, Korea, and Japan was analyzed based on EANET monitoring data. The results showed certain correlation between China's emission and the wet deposition in East Asia. The manuscript was quite well written. However, the method for trend analysis was not clear introduced. Since there are a lot of anthropogenic sources and other natural sources of Na+ than sea-salt in China, the necessity of sea-salt correction for SO42- concentration in precipitation need be reconsidered. Some detailed comments are:

In the revised manuscript, the methodology to calculate linear trend are explicitly introduced. Please refer the reply to the specific comment of 5). Regarding Na sources, we performed and checked the analysis of the correspondence between $SO_4^{2-}$ and nss-$SO_4^{2-}$ concentration in precipitation around Beijing in China. Please refer the reply to specific comments of 3) and 6).

1) Page 1, Line 15: The ratio is an index only for relative contribution of SO2 and NOx.

We have added a short note to clarify this point as follows on P1, L17:

"…to investigate the relative contributions of these acidifying species."

2) Page 3, Line 21: As I know, a long-term (2001-2013) monitoring of precipitation chemistry at one of the IMPACTS site (TSP) was recently reported by Yu et al. (ES&T, 2017).

Thank you for bringing this to our attention. We have added the suggested reference and a brief explanation of the study (e.g., P3, L22; P5, L28-29, P13, L21-22).

3) Page 4, Line 21: I suggest not to do sea-salt correction for SO42-, at least in China.

We agree that there are Na sources other than seasalt, particularly in China. Na might be affected by wind-blown dust in spring, biomass burning in autumn, and coal burning in winter. The impact of Na can be considered as minor on annual mean base because rainfall event mostly occurred during summer. The following figure shows the relation between annual mean (2008-2010) of $SO_4^{2-}$ and nss-$SO_4^{2-}$ concentration in precipitation analyzed on ten sites around Beijing in China. From this analysis, we can confirm that there are no significant differences between $SO_4^{2-}$ and nss-$SO_4^{2-}$ concentration in precipitation in China. On the other hand, in Korea and Japan, where surrounded with oceans, Na source from sea-salt would have impact. In this study, to ensure consistent data types for China, Korea, and Japan, we would like to show results for non-seasalt $SO_4^{2-}$.

[Figure]

Supplemental figure: Scatter-plot of annual mean (from 2008 to 2010) $SO_4^{2-}$ and nss-$SO_4^{2-}$ concentration in precipitation at ten sites around Beijing in China.

4) Page 4, Line 24: Is the monthly mean concentration volume-weighted?

Yes, we have explicitly indicated that volume-weighted monthly means were used in the revised manuscript on P4, L29-30.

5) Page 5, Line 4: I can not see the introduction on the method for trend analysis.

Linear trends were determined based on a linear regression analysis. We have described these methods as follows on P 8, L 24-25:

"Linear trends during each of the three phases were analyzed using linear regression, and significance levels were calculated using the Student's $t$-test."

6) Page 5, Line 32: Na+ can not be used as sea-salt tracer, at least in China.

In this study, to facilitate comparisons among values for China, Korea, and Japan, we would like to use the same data types and thereby would like to present non-seasalt $SO_4^{2-}$. Please also refer the reply 3).

7) Page 11, Line 7: Double 'Xiang Zhou'.

This was an error and should read "Zhuxiandong." In the revised manuscript, the correct site classification has been included in Table 1.

8) Page 26, Line 4: Better use equivalence as the unit for the sum.

We have revised the calculation of the sum as suggested. Figure 7 and the relevant explanation in the main manuscript have been revised.

9) Page 28 and 29, Table 2 and 3: How to calculate the trend?

The trend was estimated using linear regression. We have clarified this point in the revised manuscript. Please refer the reply 5).

W. Aas (waa@nilu.no)

This study compare observed trends in in-situ observations of sulfate and nitrate in precipitation, and satellite data of NO2 and SO2 with emission estimates of SOx and NOx in East Asia for the last 15 years. This is an important topic since there has been large changes in the emissions in Asia the last decades. The comparison between satellite data with emission, I think is good, but the comparison with in situ data I find a bit shallow. The spatial variability of the emissions and the wet deposition are large, and the site representatvity is a critical question. Are the sites in question able to show the overall trends for the different regions? For South Korea and Japan it is probably OK, but in China it is too few sites. The authors are aware of the problem and add campaign data in China, but these are only snapshots in time and cannot be used for assessing trends. I do miss more discussion of this. As I understand it, the authors claim that the ratio of NO3/SO4 is a good parameters for assessing the regional trends (better than the individual concentrations), but this is mainly showing that the chemical regime is changing, which is probably correct, maybe not all over the continent though, and does it help us to understand whether the individual trends of NO3 and SO4 are correct?

A critical point is that the paper is the merging all type of sites. In the EANET network, there are many urban sites and (some of) these may not show representative trends for the region. Even though the authors at page 6 line 8-14 discuss this a bit, and claim they (the Chinese urban sites) show similar trends, it is very few sites and only trends for the ratio is presented. I will suggest to separate urban and regional sites in the study, and even better calculate the trends at the individual sites rather than averaging all the sites, which have totally different siting criteria.

In this study, we aimed to present a comprehensive view of precipitation chemistry at the country scale for China, Korea, and Japan. As pointed out by the reviewer, EANET observation sites over China provide insight into long-term trends but are limited with respect to spatial coverage. To overcome this limitation, we combined these data with the IMPACTS data over southern China and coordinated observation campaign data around Beijing by LAPC/IAP/CAS. These data are limited with respect to the timescale and only provide snapshots; however, we believe that the comparison between these $NO_3^-$ and nss-$SO_4^{2-}$ concentrations and *Ratio* can provide important insights. By comparing EANET with these data (Figs. 2 (c) and (e); Fig. 5 (b)), we can confirm that the annual means are highly consistent among data types. Although the individual site and precipitation event might be affected by regional (or smaller scale) characteristics, precipitation chemistry will be triggered by large-scale characteristics of the annual mean. To reinforce this analysis, the separation of urban and regional (rural and remote) sites in China is important. In the revised manuscript, we have included the classification analysis in Figs. 2 (c) and (e) and Fig. 5 (b) for the $NO_3^-$ concentration in precipitation,

nss-SO$_4^{2-}$ concentration in precipitation, and *Ratio*. The correlations between mean values averaged over urban sites and rural and remote sites were 0.21, 0.78, and 0.65 for the NO$_3^-$ concentration in precipitation, nss-SO$_4^{2-}$ concentration in precipitation, and *Ratio*, respectively. The relatively low correlation for the NO$_3^-$ concentration in precipitation reflects the regional characteristics, and particularly differences in nitrogen species. A strong correlation was found between the nss-SO$_4^{2-}$ concentration in precipitation and *Ratio*; hence, we concluded that *Ratio* observed in the EANET network is a representative dataset for the precipitation chemistry in China. Please also see our responses to specific comments.

I also find the trend calculations a bit too optimistic since you calculate trends from only five years periods, usually one needs at least 7 years (preferably 10) for calculating trends. You have not described how you calculate trends and the significance, except that it is linear.

Linear trends during the three phases were estimated using linear regression, and significance levels were calculated using the Student's *t*-test. We have described this analysis in the revised manuscript on page 8, lines 17–18. Linear trends over five years were used to evaluate emission variation in each phase (Figs. 2 (b) and (d)). This time period is short, and we have established criteria to exclude outliers by carefully checking the original data. The analyses of individual sites will be interesting; however, due to the strict screening of outliers, data for some years are lacking for some sites. Therefore, our aim was to show average values for characteristics of precipitation chemistry over China, Korea, and Japan.

More specific comments to the text:

1) Page 1. The references Endo et al and Ban et al are mainly studying EANET data and not directly with US and Europe. You should rather use US and European references when comparison. Some examples:

EMEP: Tørseth, K. et al. Introduction to the European Monitoring and Evaluation Programme (EMEP) and observed atmospheric composition change during 1972-2009. Atmospheric Chemistry and Physics 12, 5447-5481, doi:10.5194/acp-12-5447-2012 (2012).

CASTNET: Sickles Ii, J. E. & Shadwick, D. S. Air quality and atmospheric deposition in the eastern US: 20 years of change. Atmos. Chem. Phys. 15, 173-197, doi:10.5194/acp-15-173-2015 (2015).

IMPROVE: Hand, J. L., Schichtel, B. A., Malm, W. C. & Pitchford, M. L. Particulate sulfate ion concentration and SO2; emission trends in the United States from the early 1990s through 2010. Atmos. Chem. Phys. 12, 10353-10365, doi:10.5194/acp-12-10353-2012 (2012).

NADP: Lehmann, C. M. B., Bowersox, V. C., Larson, R. S. & Larson, S. M. Monitoring Long-term Trends in Sulfate and Ammonium in US Precipitation: Results from the National Atmospheric

Deposition Program/National Trends Network. Water, Air, & Soil Pollution: Focus 7, 59-66, doi:10.1007/s11267-006-9100-z (2007).

UNECE also has relatively new assessments of trends from Europe and North Americ, which might be relevant as well:

http://www.unece.org/index.php?id=42861 and http://www.unece.org/index.php?id=42947

We appreciate the summary of references related to the US and European observation network. Based on these additional references, we have revised the sentence as follows on P2, L10-13:

"According to the Clean Air Status and Trends Networks (CASTNET) in the US (Sickles and Shadwick, 2015) and the European Monitoring and Evaluation Programme (EMEP) in Europe (Torseth et al., 2012), in Japan, which is located in the downwind region of the Asian continent, the total wet and dry deposition amounts have surpassed those of both the US and Europe (Endo et al., 2011; Ban et al., 2016)."

2) Page 3 line 13-16. I don't understand what you mean here that "the deposition were centred". with a reference to Pan 2015, who looks at deposition of trace elements in Northern China. I assume the authors are discussion the point I address in the beginning with representativity of the EANET sites, but I don't understand how you can state that "The approach taken here will further promote our understanding of precipitation chemistry for all of China" since the additional sites only cover a short period and cannot be used for trends. If you had looked at one specific year to assess the spatial deposition it would have been different.

We have revised the reference to Pan et al. (2012, 2013), which discusses sulfur and nitrogen deposition based on ten coordinated observations around Beijing. We agree that the additional datasets for southern China and northern China were limited with respect to the time period; however, in combination with the EANET observations over four regions, we believe that it is possible to characterize the precipitation chemistry over China. We have explained this point in the revised text as follows on P3, L15-17:

"Although these additional data over southern and northern China are limited to one year or a few years, this approach of combining and comparing observational datasets will further improve our understanding of precipitation chemistry over China, broadly."

Considering the representativeness of EANET sites and the reviewer's comment, we have added analyses in which ten EANET sites over China, of which we classified as urban sites (n = 6) and rural and remote sites (n = 4).

3) Page 4 line 6. The Ogasawara site was excluded with quite strict criteria. 25% is more appropriate. Further you could have in included the Russian site close to the Korean border, Primorskaya, which is also downwind from the large emission sources in China.

Because the Ogasawara islands are located on the Pacific Ocean, values below the detection limit were found, namely, 11.8% for $NO_3^-$ and 7.7% for nss-$SO_4^{2-}$. For other observation sites in Japan, values are typically over the detection limit. Hence, we have excluded the Ogasawara site from this study.

Thank you for your comment regarding the site in Russia. In fact, Primorskaya showed similar trends to those observed in China, Korea, and Japan as is illustrated in the figure below. Analyses of the Russian site are beyond the scope of this study; hence, we prefer not to include these results in the manuscript.

[Figure]

Supplemental figure: Long-term temporal variation in (a) precipitation, (b) $NO_3^-$ concentration in precipitation, (c) nss-$SO_4^{2-}$ concentration in precipitation, and (d) *Ratio* are illustrated in the figure below.

4) Page 8 line 13-14: "For the treatment of precipitation amount, months where data were insufficient were the same as when applying the Smirnov-Grubbs test for Ratio calculation." I don't understand this sentence

We have revised the sentence as follows and moved to section 2.1 on P5, L11-14):

"For the treatment of the monthly precipitation amount, the Smirnov–Grubbs test for *Ratio* was used to discard outliers. If at least 9 months were available, the annual accumulated precipitation amount was calculated."

5) Page 8 line 16-17 The sentence "Statistical analysis revealed that, except for the increasing and decreasing trend over China and Korea during Phase III .. there was no clear change in precipitation amount..." is somewhat in contrast to the conclusion later on page 13 line 21: "In spite of the increasing trends of precipitation amount, decreasing trends for nss-SO4 wet deposition amounts over China, Korea, and Japan were seen after 2005–2006". The increase in precipitation amount in China is after 2010 (though the variability is very high) and South Korea has decreasing amount.

We apologize for the apparently conflicting statements. We have carefully revised the conclusion as follows on P14, L9-12:

"Despite the increasing trends in the precipitation amount, decreasing trends for nss-SO$_4^{2-}$ wet deposition over China were observed after 2005–2006 and matched reductions in SO$_2$ emissions from China. This decrease in China triggered the decreases in nss-SO$_4^{2-}$ wet deposition over Korea and Japan."

6) Page 8-9 "The temporal variation found in the NO3 concentration in precipitation did not correspond to the NOx emissions variation". This non-linearity can be due to several factors. E.g.: The non representativity of the sites, the change in atmospheric composition and chemical regimes (changes in base cations and ammonium), oxidation capacity of the atmosphere, all may change the lifetime of NOx and NO3 (course or fine).

We have added potential explanations for the nonlinearity as follows on P9, L20-22:

"This may be explained by changes in atmospheric composition and chemical regimes with changes in base cations and the oxidation capacity of the atmosphere, leading to changes in the lifetime of nitrogen species."

7) Page 9 line 18. "Contamination". It's a bit misleading word. The satellite measurements are not contaminated they are influence by sources outside Korea (like China)

We have replaced "contamination" with "impacted" on P10, L7.

8) Page 9 line 23. Have the volcanic activity changed during the period to influence the trend of the SO2 column data? Should maybe also have been included also in the emission inventories?

According to the volcanic activity report by Japan Meteorological Agency (http://www.data.jma.go.jp/svd/vois/data/tokyo/volcano.html,), the activity above south Kyushu (a western island of Japan) increased from Phase II to Phase III. This can be detected in satellite observations (Fig. 4), and the activity of Miyakejima, located south of Tokyo, gradually decreased after the eruption in 2000. As we have stated on P10, L32-34, relatively

high nss-$SO_4^{2-}$ concentrations in precipitation in Japan in 2001–2002 were partially related to the activity of Miyakejima.

9) Page 9-10. IMPACTS sites have lower levels compared to EANET since these are mainly regional/rural sites while the BNU site is urban and naturally higher than the average EANET site.

We have clarified these points as follows on P9, L25-28:

"When comparing EANET observations with additional datasets, IMPACTS sites around south China showed lower $NO_3^-$ concentrations in precipitation, reflecting the characteristics of remote sites, while BNU and LAPC/IAP/CAS sites around Beijing showed relatively high concentrations, reflecting the emission intensity over northern China."

10) Page 10: There are some contradicting statements:

1) Over China, NO2/SO2 column ratio were flat during Phase I, sharply increasing during Phase II, and almost flat during Phase III.

2) NOx/SO2 emission ratio were well correlated with the variation in Ratio over China

3) Ratio was almost 0.3 during Phase I and subsequently increased.. during Phase II, with a trend of +14.8±1.9%/year and around 0.4–0.6 during Phase III with a trend of +10.1±3.8%/year (p < 0.05).

This leads to the conclusion on page 11. "Ratio observed in EANET network can be a representative dataset of China for the precipitation chemistry" This is not obvious for me

By separating the EANET observation sites, we clearly observed that the mean values for urban sites and for rural and remote sites were highly correlated (correlation coefficient, 0.65). The manuscript has been revised for clarity as follows on P11, L20-32:

"*Ratio* was almost 0.3 during Phase I and subsequently increased to 0.5 during Phase II, with a trend of +14.8 ± 1.9 %/year (p < 0.001), and to approximately 0.4–0.6 during Phase III, with a trend of +10.1 ± 3.8 %/year (p < 0.05). The analyses of different site types for EANET observations over China (Fig. 5 (b), see Table 1 for classification) revealed that the mean values for urban sites and rural and remote sites were highly correlated, with a correlation coefficient of 0.65. In addition to the variation of *Ratio* based on EANET observations for urban sites and rural and remote sites, other observations through IMPACTS, BNU, and LAPC/IAP/CAS also indicated similar variation in *Ratio*. Based on the IMPACTS dataset, which covered southern China with extensive data in 2001–2003, *Ratio* in China during Phase I was approximately 0.2, which was much lower than the current (Phase III) status. Although $NO_3^-$ and nss-$SO_4^{2-}$ concentrations in precipitation were higher than those in the EANET observation dataset, observations at BNU revealed a *Ratio* of 0.31, which was consistent with EANET observation results. A coordinated observation network by LAPC/IAP/CAS, which

operated for three years around Beijing, indicated an increasing trend during Phase II from 0.39 in 2008 to 0.59 in 2010. These results obtained from multiple observations reinforce the idea that *Ratio* observed in the EANET network is representative of the precipitation chemistry in China."

11) Page 12 line 31. Nr. Is this reactive nitrogen? It is not defined. Can also be interpret as reduced nitrogen. I assume reactive nitrogen since you at page 13 line 1 states that Nr can cause eutrophication and this means both NO3 and NH4

Nr refers to reactive nitrogen; it has been defined on P13, L1.

12) Figure 7c. It is not clear how sum NO3 and SO4 is calculated Figure C do not add up from fig a and b, even if changing units. Maybe better to look at equivalent and not mass.

We have added a brief explanation of the calculation used in Figure 7 (c). Considering the suggestion, the revised manuscript shows the equivalent basis (keq/ha) for the sum of $NO_3^-$ and $SO_4^{2-}$, and we have recalculated the $NO_3^-$ fraction on an equivalent basis.

13) Table 2. It would have been nice to indicate how many sites are included in these calculations, and the time period for the different phases should be included in the table so the reader don't need to search in the text. Is it the same number sites in all the phases? In the table there is no unit (kg/ha pr year?)

We have indicated the total numbers used to calculate the means and trends in Tables 2 and 3. The time periods are also included in Tables 2, 3, and 4. The methods used to calculate the linear trends and the units are included in the note.

[revised manuscript text omitted]